# PICTUREE—Aedes: A Web Application for Dengue Data Visualization and Case Prediction

**DOI:** 10.3390/pathogens12060771

**Published:** 2023-05-29

**Authors:** Chunlin Yi, Aram Vajdi, Tanvir Ferdousi, Lee W. Cohnstaedt, Caterina Scoglio

**Affiliations:** 1Department of Electrical and Computer Engineering, College of Engineering, Kansas State University, Manhattan, KS 66506, USA; cyi@ksu.edu (C.Y.); avajdi@ksu.edu (A.V.); tanvirf@ksu.edu (T.F.); caterina@ksu.edu (C.S.); 2National Bio- and Agro-Defense Facility, Agricultural Research Service, United States Department of Agriculture, Manhattan, KS 66502, USA

**Keywords:** web application, dengue transmission, mosquito population estimation, dengue risk estimation, dengue incidence forecasts

## Abstract

Dengue fever remains a significant public health concern in many tropical and subtropical countries, and there is still a need for a system that can effectively combine global risk assessment with timely incidence forecasting. This research describes an integrated application called PICTUREE—Aedes, which can collect and analyze dengue-related data, display simulation results, and forecast outbreak incidence. PICTUREE—Aedes automatically updates global temperature and precipitation data and contains historical records of dengue incidence (1960–2012) and *Aedes* mosquito occurrences (1960–2014) in its database. The application utilizes a mosquito population model to estimate mosquito abundance, dengue reproduction number, and dengue risk. To predict future dengue outbreak incidence, PICTUREE—Aedes applies various forecasting techniques, including the ensemble Kalman filter, recurrent neural network, particle filter, and super ensemble forecast, which are all based on user-entered case data. The PICTUREE—Aedes’ risk estimation identifies favorable conditions for potential dengue outbreaks, and its forecasting accuracy is validated by available outbreak data from Cambodia.

## 1. Introduction

The PICTUREE—Aedes web application is an innovative system that integrates tools for displaying dengue-related data, visualizing simulated results, and forecasting future cases for an ongoing outbreak. This application is a product of the Predicting Insect Contact and Transmission Using histoRical Entomological and Environmental data (PICTUREE) project, which was initiated in 2019.

Dengue fever is an *Aedes* mosquito-borne viral infection that threatens the health of over 3 billion people in numerous tropical and subtropical regions [1]. *Aedes aegypti* mosquito is the primary carrier of dengue viruses, with the ability to bite multiple individuals in a brief period, and its immature phase can be found in water containers near human dwellings [2]. The proliferation, biting rates, infective rate, and survivability rate of *Aedes* mosquitoes are substantially affected by climatic conditions such as rainfall, temperature, and relative humidity, which, in turn, have a significant impact on the spread of the dengue virus [3,4,5]. The breeding time and maturity period of *Aedes* mosquitoes are shortened due to global warming, resulting in the accelerated growth of the vector population [6]. Furthermore, increasing human mobility and high population density have facilitated the geographical spread of dengue viruses over long distances [7]. Therefore, an integrated system is needed to aggregate collected data, analyze them, estimate parameters, and forecast future risks and incidence to proactively combat the spread of the virus.

Numerous research studies have been conducted to identify and quantify the temperature-dependent variables that impact the life cycle of the mosquito vector, due to the susceptibility of the mosquito’s life cycle to temperature [8,9,10,11]. Mathematical equations can be used to model the dynamics of the mosquito population and to estimate the parameters such as mortality, transition rate, and oviposition rate at different temperatures [5]. Another dengue transmission model, which took into account various temperatures, survival rates, between-bite intervals, and extrinsic incubation periods, discovered that an increase in mosquito longevity and temperature led to an augmentation in dengue transmission [12].

There has been an increasing number of approaches to predicting the incidence of infectious diseases, including the use of statistical models [13,14], state-space compartmental models [15,16,17,18], deep learning methods [19,20,21], etc. The state-space models coupled with data assimilation algorithms can generate real-time forecasts and estimate key parameters. The estimated model parameters can then be utilized to determine the basic reproduction number of the disease, the carrying capacity of the vector population, and other relevant metrics. A state-space compartmental disease model combined with an ensemble Kalman filter (EnKF) was employed to estimate and predict future dengue incidence, infection rates, and mosquito emergence rates in [16]. Deep learning approaches, especially the long short-term memory (LSTM) recurrent neural network, are also widely used to generate reliable epidemic forecasts [21]. Machine learning approaches have the advantage of being able to handle nonlinearity and complex data structures. Another emerging technique is the ensemble forecast that enhances the forecasting accuracy of individual models [22,23]. A superensemble forecast was developed by combining three models—susceptible, infectious, recovered, ensemble adjustment Kalman filter, Bayesian weighted outbreaks, and historical likelihood—using the weighted average method [23]. The superensemble was used to predict dengue cases in San Juan, and it was found to provide more accurate forecasts compared to individual models. The use of ensemble forecasts, which combine multiple forecasting models or techniques, can help reduce errors and biases that may be present in individual forecasts.

The PICTUREE—Aedes web application aggregates dengue-related data from multiple sources, including global daily temperature, global daily precipitation, *Aedes* mosquito occurrence, and dengue case occurrence. These data are stored in an SQL database and can be readily visualized through user-friendly displays. The PICTUREE—Aedes web application also provides estimations of the *Aedes* mosquito population, the dengue transmission reproduction number $R_{t}$, and the dengue risk globally, which can be viewed by users in both space and time. Furthermore, the application employs several algorithms, including the ensemble Kalman filter, the recurrent neural network, the particle filter, and the super ensemble, to forecast dengue cases for an ongoing outbreak. To generate forecasts, users must provide information about the cumulative reported cases in recent weeks, the outbreak location, and the local population size. After processing this data, the application outputs the statistics of the three-week forecasted cases.

## 2. Data

Table 1 presents a comprehensive list of the data sources. The daily global temperature and precipitation data are obtained from the National Oceanic and Atmospheric Administration (NOAA) weather stations [24]. The ERA5 hourly data provide hourly estimates of atmospheric variables, such as temperature, pressure, and humidity, which are used for simulation purposes [25]. The terrestrial ecoregions data are utilized to select areas with similar climate conditions [26,27,28]. The *Aedes* mosquito occurrence dataset and dengue occurrence dataset are geo-positioned occurrence records spanning from 1960 to 2012 [29,30]. Country-wise Google trend data for the keywords “dengue” and “fever” are also collected. The aforementioned data are either directly displayed on the map or are used for the simulations and forecasts [31].

## 3. Simulation Tools

The PICTUREE-Aedes simulations consist of two primary components: (1) the estimations of dengue risk-related measures and (2) the forecasts of case number for an ongoing outbreak. The tools graphically represent current and future risk of dengue transmission for users based on the life history of the mosquito vector.

### 3.1. Estimating Dengue Risk-Related Measures

We developed a dengue transmission risk model by training a neural network classifier over historical dengue outbreak data. The model uses the estimated vector population and the Dengue transmission reproductive number as predictive variables. In order to estimate the vector population and reproductive number, we developed a mechanistic model to simulate the *Ae. aegypti* population. The mechanistic model incorporates the mosquito’s temperature-dependent parameters in a system of integral equations to calculate the mosquito population at various stages of development, including immature and adult stages. A diagram of the mechanistic model is shown in Figure 1. The set of integral equations precisely considers the temperature-dependent development time required for mosquitoes to transform from an egg into adult form. Given that the development period tends to be longer in seasons with cooler temperatures, such precise calculation is especially crucial in estimating the mosquito population in areas that experience significant temperature fluctuations throughout the year.

#### 3.1.1. Vector Population

To accurately simulate the mosquito population fluctuations, the model accounts for the time spent in each life history stage during the aquatic or immature portion of the mosquito life cycle, as shown in Figure 1. The development rates of these stages are temperature-dependent, with warmer temperatures resulting in faster development. The temperature-dependent rates were determined from experimental data [5,8,10]. In addition to temperature, the effect of precipitation on the mosquito population is increased through the carrying capacity of the location: more rainfall equates to greater availability of habitats or more containers filled with water. Alternatively, less rainfall in some urban areas results in more irrigation or water storage, leading to an increase in filled containers, but this is location dependent. The equations that describe the life cycle of the mosquitoes are non-Markovian, meaning that transition times to new stages in the cycle are not exponential random variables. The transition to a new stage happens when the development of a current stage is complete. Moreover, the development rates are strongly temperature-dependent, and temperature changes throughout the year. Therefore, we use integral equations to describe the life cycle of the mosquitoes accurately.

In order to estimate the impact of precipitation on carrying capacity, the life cycle equations are extended to account for virus transmission and to subsequently fit the model to various dengue outbreak data from around the world using particle filter smoothing. This enables us to indirectly estimate the carrying capacity and its variation with precipitation. For this process, we used 67 dengue epidemic curves from Brazil, Mexico [32], Malaysia [33], Pakistan [34], Philippines [35], Singapore [36], Taiwan [16], Sri Lanka [37], Costa Rica [38] and Vietnam [39]. Particle filtering is an estimation technique known as sequential Monte Carlo, which involves iteratively updating the estimated parameters using a sequence of observation. We used a number of new cases as observations, and the carrying capacity is the unknown parameter that is sequentially estimated.

The local mosquito populations are calculated using the life cycle equations, daily average temperature, and precipitation for each location around the globe with longitude and latitude resolutions of 0.25 degrees. The result is the time-dependent profile of the mosquito population. The calculated values are not the exact populations, rather they reflect the variation of the mosquito population in that location through time.

#### 3.1.2. Dengue Transmission Reproductive Number

The value of dengue’s basic reproductive number $R_{0}$
(1)$$R_{0}=\frac{V}{N^{H}}\frac{\alpha^{2}\phi^{H\to V}\phi^{V\to H}\gamma^{V}}{\mu^{V}\sigma^{H}\left(\mu^{V}+\gamma^{V}\right)}$$
depends on the ratio of mosquito to human population $V/N^{H}$, mosquito biting rate α, human-to-vector and vector-to-human transmission probabilities per infectious bite $\phi^{H\to V},\phi^{V\to H}$, extrinsic incubation period in the mosquito $\gamma^{V}$, mosquito mortality rate $\mu^{V}$, and infectious period of human $\sigma^{H}$. These parameters are temperature-dependent and can be found in [40], except for $V/N^{H}$. The $R_{0}$ is derived assuming constant temperature and using Markovian equations. When $R_{0}<1$, the viral transmission will result in fewer new cases than past cases, and the infection will burn out if maintained for sufficient time. Conversely, interpretation that the infection is going to die out when $R_{0}<1$ is only relevant if this condition is satisfied for a long period. However, if the temperature is almost constant for a period longer than the relevant periods in the transmission model, the $R_{0}$ can be used to quantify the severity of virus transmission. In other words, when $R_{0}>1$, an infected individual is expected to generate more infection in the population. For this reason, we utilize the value of $R_{0}$ to gauge the severity of virus transmission. The mosquito-to-human population ratio $V/N^{H}$ is time-dependent and is affected by the carrying capacity of the environment. To estimate the carrying capacity per human, we fit the transmission model to epidemic dengue curves from different locations around the world with a given human population. For each location, we obtain a constant component for the carrying capacity and a parameter that determines the variation of the carrying capacity with precipitation. We use the distribution of these parameters to find a statistical model for the carrying capacity per human. Although this model may not give the exact value for the carrying capacity, we use it as an approximate input in the calculation of $R_{0}$. Our calculation shows that $R_{0}$ is almost proportional to the carrying capacity; therefore, the calculated $R_{0}$ provides a good approximation for the time-dependent profile of the exact reproductive number. Higher $R_{0}$ values indicate greater dengue virus transmission in an area; therefore, users can quickly understand where control measures need to be initiated and evaluate if they are working by observing changes in $R_{0}$.

#### 3.1.3. Dengue Transmission Risk

To estimate dengue transmission risk, we use the dengue transmission reproductive number and vector population calculated using the mechanistic model as predictive variables. For each location, we calculate the one-week moving average of the variables and apply them in a trained neural network classifier to obtain a time-dependent transmission risk. Assuming the historical presence of the mosquitoes and the virus, the risk factor is a value between zero and one corresponding to the least and the most favorable environmental condition, respectively, for dengue transmission. To train the model, we used dengue epidemic data from Brazil, Mexico, Malaysia, Pakistan, Philippines, Singapore, Taiwan, Sri Lanka, Costa Rica and Vietnam. For each one of the epidemic curves, we manually tagged periods of high and low transmission. If the number of new cases is small and comparable to the baseline for the location, we tag that period as zero, otherwise as one. In addition, we calculate the corresponding predictive variable as described above for each location and period. Using the tag values as the target variable and the corresponding predictive variables, we train the neural network classifier for risk evaluation.

In Figure 2, we show the data used for training as green and yellow dots, indicating low (0) and high (1) transmission, respectively. The neural network classifier is a nonlinear model that assigns a risk value between zero and one to points on the surface defined by the predictive variables $V/N^{h}$ and $R_{0}$. Within the depicted image, regions displaying risk values exceeding 0.5 are indicated by the color red, while those with values below 0.5 are represented by the color blue.

### 3.2. Forecasting Dengue Case Number

In this section, we outline the methodology we have developed for predicting the future incidence of ongoing dengue outbreaks. Our approach entails analyzing user-provided data, including reported incidence, location, and date, and extracting pertinent environmental information from our extensive database. These data are then utilized to generate forecasts, which are presented by the statistics including upper quantile, median, and lower quantile values of the forecast distribution. Case forecasts help PICTUREE–Aedes users understand how the dengue outbreak will change with time. Forecasts of increasing cases suggest that allocating addition resources to an area is needed due to continued transmission, whereas decreasing case forecasts suggest that resources can be reallocated and that the outbreak is getting smaller.

#### 3.2.1. SEIR-SEI-EnKF Forecast

The SEIR-SEI-EnKF forecasts are generated by employing the disease transition model SEIR-SEI combined with the EnKF data-assimilation technique. The SEIR-SEI-EnKF model is a new approach for estimating and forecasting dengue outbreak dynamics, which has been detailed in [16]. The SEIR-SEI model is a compartmental model that describes the transmission of the dengue virus between the host and vector population groups. This model considers the compartments for the human population and vector population, such as susceptible ($S^{H}\;and\;S^{V}$), exposed ($E^{H}\;and\;E^{V}$), infectious ($I^{H}\;and\;I^{V}$), and removed ($R^{H}$). Transitions between compartments are described by the following difference equations
(2)$$X_{t+1}=f\left(X_{t}\right)=\left\{\begin{array}{c} S_{t}^{H}+\lambda^{H}P_{t}^{H}-\beta^{V\to H}S_{t}^{H}I_{t}^{V}-\mu^{H}S_{t}^{H}\\ E_{t}^{H}+\beta^{V\to H}S_{t}^{H}I_{t}^{V}-\left(\delta^{H}+\mu^{H}\right)E_{t}^{H}\\ I_{t}^{H}+\delta^{H}E_{t}^{H}-\left(\gamma^{H}+\mu^{H}\right)I_{t}^{H}\\ R_{t}^{H}+\gamma^{H}I_{t}^{H}-\mu^{H}R_{t}^{H}\\ S_{t}^{V}+\lambda^{V}P_{t}^{V}-\beta^{H\to V}S_{t}^{V}I_{t}^{H}-\mu^{V}S_{t}^{V}\\ E_{t}^{V}+\beta^{H\to V}S_{t}^{V}I_{t}^{H}-\left(\delta^{V}+\mu^{V}\right)E_{t}^{V}\\ I_{t}^{V}+\delta^{V}E_{t}^{V}-\mu^{V}I_{t}^{V} \end{array}\right.$$
where $X_{t}={\left[S_{t}^{H},E_{t}^{H},I_{t}^{H},R_{t}^{H},S_{t}^{V},E_{t}^{V},I_{t}^{V}\right]}^{⊺}$, P,λ,μ,β,δ,and γ represent the population, birth rate, death rate, infection rate, incubation rate, and recovering rate, and upper note *H* for host species, and *V* for vector species.

The Ensemble Kalman Filter (EnKF) is a data assimilation technique that can be used to estimate both the states and parameters of a system simultaneously. The EnKF maintains an ensemble of the state variables $X_{t}$ and the time-varying parameters $\Phi_{t}={\left[\beta^{V\to H}\beta^{H\to V}\lambda^{V}\right]}^{⊺}$, which can be put together in a matrix $A_{t}\in R^{10\times N}$ where *N* is the number of ensembles
(3)$$A_{t}=\left[\begin{array}{cccc} X_{t}^{1} & X_{t}^{2} & \cdots & X_{t}^{N}\\ \Phi_{t}^{1} & \Phi_{t}^{2} & \cdots & \Phi_{t}^{N} \end{array}\right].$$

At each time step, the EnKF first uses the model simulations to predict the prior, which is the expected state of the system
(4)$$A_{t+1}^{-}=F\left(A_{t}\right)=\left\{\begin{array}{c} f(X_{t})\\ g(\Phi_{t}) \end{array}\right.+q_{t}$$
where $g(\Phi_{t})$ is the difference equation for the parameters, and $q_{t}$ is the model noise. Therefore, the ensemble covariance for the prior is given by
(5)$$\sigma_{t}^{pri}=\frac{1}{N-1}\left(A_{t}^{-}-\bar{A_{t}^{-}}\right){\left(A_{t}^{-}-\bar{A_{t}^{-}}\right)}^{⊺},$$
where $\bar{A_{t}^{-}}$ is the mean of ensemble members.

To update the estimated state and parameters of the system, the EnKF assimilates new observations $y_{t}$ into the ensemble $\hat{Y_{t}}=y_{t}+\epsilon_{t}$, where $\epsilon_{t}$ is the perturbation vector. The assimilation step involves comparing the predicted states of the ensemble with the observed data and adjusting the estimated state and parameters of the system to better fit the observations
(6)$$A_{t}=A_{t}^{-}+K_{t}\left(\hat{Y_{t}}-H_{t}A_{t}^{-}\right)$$
where $K_{t}=\sigma_{t}^{pri}H_{t}{\left(H_{t}\sigma_{t}^{pri}H_{t}^{⊺}+\sigma_{t}^{obs}\right)}^{-1}$ is the Kalman gain, and $H_{t}$ is the measurement index matrix, $\sigma_{t}^{obs}=\frac{\epsilon_{t}\epsilon_{t}^{⊺}}{N-1}$. Finally, the EnKF generates a new ensemble of model simulations by perturbing the updated estimated state and parameters of the system, and the process repeats for the next time step. In summary, the EnKF utilizes a data assimilation approach that combines real-time observations with mathematical models to accurately forecast dengue epidemics, helping to inform proactive measures for disease control and prevention.

#### 3.2.2. Neural Network Forecast

The second approach to predict weekly cases uses a recurrent neural network model, as shown in Figure 3. Assuming that the index of the current week is represented by *t*, the model uses a sequence of weekly new cases up to *t*, $c_{t-4},c_{t-3},c_{t-2},c_{t-1},c_{t}$, as input to predict new cases in the next three weeks. In addition to the sequence of new cases, the model uses the vector population and reproductive number sequences, calculated independently using the model discussed in the previous section.

The vector population $V_{t+h-4},V_{t+h-3},V_{t+h-2},V_{t+h-1},V_{t+h}$ and reproductive number $\left\{R_{t+h-4},R_{t+h-3},R_{t+h-2},R_{t+h-1},R_{t+h}\right\}$ include estimated vector population and reproductive number in the weeks ahead. Since the calculation of these sequences requires temperature as input, we use the average of temperature for the corresponding period in the past years. In addition to the sequences we discussed, we feed the error of the latest two predictions to the neural network model. We train this neural network model using historical dengue epidemic curves from different locations around the world. Although it is possible to train a neural network model that generates one, two, and three weeks of predictions, we choose to train independent models for each one of these prediction horizons. Here, the obtained output is a single value for each prediction time interval.

#### 3.2.3. Particle Filter Forecast

As a third approach, we apply particle filter prediction to the transmission model, as depicted in Figure 4. In the transmission model, we divide the adult female mosquitoes into three compartments based on the virus development in the mosquitoes. We assume that all the newly developed adult mosquitoes are virus-free, (Adult-S). During the gonotrophic period, whose length is temperature-dependent, the mosquitoes may bite an infectious human, acquire the virus, and remain in the adult-exposed (Adult-E) state until they become infectious and are able to transmit the virus. The sojourn time for the Adult-E state is temperature-dependent. After becoming infectious, the mosquitoes infect susceptible humans during blood meals in their gonotrophic cycle until they die. We assume that the number of infecting bites per cycle is constant, and the bites occur uniformly in each cycle period. The human infection rate is $A_{I}B/\left(CN\right)$, where $A_{I}$ is the population of infectious mosquitoes, *B* is the number of infecting bites per gonotrophic cycle, *C* is the cycle length, and *N* is the human population. Similarly, the mosquitoes’ infection rate is $H_{I}B/\left(CN\right)$, where $H_{I}$ is the infectious human population. In the particle filter estimation, we assume two unknown parameters, which are carrying capacity and the number of infecting bites per gonotrophic cycle. Like the mosquito life cycle model as shown in Figure 1, we account for non-Markovian transitions in the virus transmission model. Particle filtering is a sequential Monte Carlo approach that employs a group of samples to represent the posterior distribution of the unknown parameters. In each step of the process, a weight is assigned to each sample based on the observed data. During the resampling phase, particles with low weights are substituted by new particles in close proximity to the particles with higher weights. An advantage of the particle filter is its ability to handle observations from any probability distribution. In our particle filter estimation, we provide the algorithm with a sequence of weekly new human cases and estimate posterior distributions of new human cases for the one-week up to three-week horizon.

#### 3.2.4. Super Ensemble Forecast

Using the competing forecasts described above, a super-ensemble forecast can be created by applying a weighted average of the individual forecasts. The weights of the three individual forecasting methods is determined based on the root-mean-square error (RMSE) of those forecasts over prior predictions. Specifically, the RMSE of the forecasts made for time t−3,t−2, and t−1 can be used to generate the super-ensemble forecast for time T. The RMSE of each individual forecast $e^{i}\left(i=1,2,3\right)$ can be calculated as follows: (7)$$e_{t}^{i}=\sqrt{\frac{1}{3}\sum_{\tau =t-3}^{t-1}{\left(f_{\tau }^{i}-O_{\tau }\right)}^{2}},$$
where $f_{t}^{i}$ is the median forecast of method *i* at time *t*, and $O_{t}$ is the observation at time *t*. The weight of method *i* can be obtained by
(8)$$w_{t}^{i}=\frac{1}{e_{t}^{i}\sum_{k=1}^{3}\frac{1}{e_{t}^{k}}}.$$The super-ensemble forecast is given by
(9)$$f_{t}^{se}=w_{t}^{1}f_{t}^{1}+w_{t}^{2}f_{t}^{2}+w_{t}^{3}f_{t}^{3}.$$

## 4. System Architecture

The web application consists of several components as shown in Figure 5, which we broadly categorize into (1) the frontend client application and (2) the server backend.

### 4.1. Client Application

The web-based client is built on top of HTML and is accessible using any web browser. The application’s graphical interface consists of an interactive map viewer and tools for data selection, filtering, parameter inputs, and submission of on-demand analyses. The map is displayed using the open-source JavaScript library leaflet.js [41]. The application communicates directly with the server backend to load maps, results, and data.

### 4.2. Server Backend

HTTP server: The backend system exposes multiple API (application programming interface) endpoints to the client via an HTTP server. This HTTP server runs in the Node.js environment [42].Database: The database service consists of PostgreSQL and PostGIS [43] extensions to handle spatial queries. The database contains multiple tables to store data collected from various sources and simulation results.Map tile service: When the client requests the home page of the map tool, the HTTP server loads map tiles from a separate map tile service called MapTiler [44]. This service helps with memory management by loading only the parts and zoom levels required by the client. The service takes input from large tile files collected from OpenStreetMap [45].Simulation tools: The system can deploy and run executable code to perform on-demand analysis and simulations on selected data. The HTTP service directly communicates with the executables to pass arguments and to collect results.Scheduled updates: The system contains scripts that run regularly to update the database tables with new weather data. The script connects to the NOAA-NCEI [24] servers for daily weather observation data.

## 5. Results

### 5.1. Data Visualization

Figure 6 displays the graphical interface of the map tool, where the 2D projection map of the Earth is centrally located and where users are able to pan and zoom to their areas of interest. The interface’s left side features various tools that assist with data selection and visualization. A comprehensive guide to the description and usage of each tool is available in the PICTUREE–Aedes Users Guide in the Appendix A.

The PICTUREE–Aedes enables users to identify weather stations and their recorded data based on their preferences. Figure 7a illustrates the distribution of weather stations in Central Indochina (Ecoregion). Users can select a station and plot its precipitation and temperature data. Figure 7b demonstrates the daily temperature and precipitation data of station UBON RATCHATHANI throughout 2022.

Figure 8 presents the occurrence records of dengue, *Aedes aegypti*, and *Aedes albopictus*, marked, respectively, with red, blue, and purple dots in Central Indochina. Clicking on a dot will generate a popup that displays the country name and the year of occurrence.

Additionally, the PICTUREE–Aedes supports the display of information on Google Trends. Figure 9 illustrates the trend of “dengue” and “fever” in Cambodia during January 2018.

### 5.2. Simulation Results

The PICTUREE–Aedes offers four types of mosquito simulation results, including daily average temperature, vector host ratio, reproduction number, and dengue risk, which can be visualized spatially and temporally. To view the data spatially, users must select a specific day, and a heat map of the selected area will display the selected data value. Figure 10 displays the average temperature (a), dengue transmission reproduction number (b), vector–host ratio (c), and dengue transmission risk (d) of the Indochina peninsula on 15 September 2022. Visualizing data beyond the next six months from the current time is limited.

To visualize data temporally, users must select a location and a time range. Figure 11 displays mosquito simulation data from 1 January to 11 June 2022 at the user-specified location. Please note that the query time frame is limited to a six-month horizon from the current date.

The PICTUREE–Aedes provides three types of predictive tools and their ensemble forecast. Figure 12 illustrates the predicting results of the dengue outbreak in Cambodia during 2022. The first 13 data points (before 27 April) represent weekly dengue cases, while the remaining points (including/after 27 April) denote the predictions. Figure 12a displays the distribution of SEIR-SEI-EnKF predictions for the next three weeks, including quantiles and median. Figure 12b illustrates the neural network prediction for the next three weeks. Figure 12c shows the results of the particle filter, presenting the distribution for the next four weeks. Figure 12d displays the result of the super-ensemble, showcasing the median prediction from each member, the average, and weighted average of the members.

Table 2 displays the outcome of the prediction as well as the reported cases. All of the reported cases in the following weeks are situated within the prediction range, which confirms the precision of the prediction tools.

## 6. Discussion

Dengue fever is a viral mosquito-borne disease transmitted predominately by the *Aedes* mosquitoes. It infects an estimated 400 million people annually, with nearly half the world’s population at risk of infection [46]. Dengue cases are influenced by complex interactions of ecology, environment, meteorological factors, and virus factors (serotype), among many others [47].

The PICTUREE-Aedes is an innovative and unique system that serves as a valuable resource for both researchers and health officials in their efforts to advance scientific research and to effectively manage dengue outbreaks. By collecting and aggregating a comprehensive set of critical factors related to dengue transmission, such as reanalyzed temperature, precipitation, occurrences of *Aedes* mosquitoes, and dengue cases, the PICTUREE-Aedes provides researchers with a rich dataset for their investigations. This data-driven approach allows researchers to gain deeper insights into the complex dynamics of dengue transmission and aids in the development of more effective prevention and control strategies. In addition to its research-oriented features, the PICTUREE-Aedes offers essential information to health officials, enabling them to proactively prepare and adapt to the changing situation during dengue outbreaks. The system employs sophisticated algorithms to estimate *Aedes* mosquito abundance, dengue transmission reproduction number, and dengue risk, providing crucial insights into the intensity and geographical spread of the disease. Armed with this knowledge, health officials can make informed decisions about when and where to implement targeted vector control measures, effectively minimizing the risk of dengue transmission and safeguarding public health. Moreover, the PICTUREE-Aedes goes beyond monitoring and estimation by offering valuable and accurate forecasting capabilities. By utilizing its advanced modeling techniques, the system can predict future dengue case counts, empowering health officials and healthcare facilities to deploy medical personnel and to allocate treatment resources more efficiently. This forecasting functionality is instrumental in anticipating the potential impact of dengue outbreaks, enabling proactive measures to be taken in terms of resource allocation, capacity planning, and public health awareness campaigns.

The selection of a 6-month forecast horizon was made without a specific rationale. The forecast horizon relies on the availability of dependable environmental variables for the future. Presently, we employ the average temperatures from previous years for the corresponding timeframe in the next six months. However, the current values of the calculated variables are adjusted based on the most recent available environmental data. The current levels will influence the future levels of the calculated variable. However, for the dengue case number forecast horizon, we chose 3 weeks because human intervention can change the underlying physical transmission process in the long term.

Concerning possible limitations of the proposed system, the temperatures displayed are collected by the weather stations; thus, they do not always reflect temperature in areas close to the weather station. For example, shaded and barren areas near the same weather station will be considered at the same daily temperature. In contrast, the actual temperature of those places can range significantly. There are many water sources beyond precipitation, and *Aedes* mosquitoes are very efficient at finding these areas to lay eggs, such as domestic or industrial wastewater, sprinkler systems, air conditioning units, etc. [48]. Therefore, the accuracy of the local temperature and precipitation and the estimation results obtained considering only temperature and precipitation can be erroneous in some areas, and local data are needed to adjust the models. Furthermore, human behaviors cannot be accounted for at this time [49]. Water collection during the dry seasons, litter accumulating water, and irrigating lawns have all been demonstrated to impact *Ae. aegytpi* numbers, and this global model cannot account for local behaviors at this time [50]. In the future, local models can be generated and placed within the global model. Additionally, there is an increasing significance of female mosquito vertical transmission through the dry quiescent egg population. These infected eggs play a potential role in triggering dengue incidence following periods of low mosquito activity, such as the dry season [51]. In our future work, we plan to explore the dynamics of female mosquito vertical transmission in greater depth. This will involve investigating the factors influencing the rate of infected eggs, studying the survival and viability of infected offspring, and analyzing the potential amplification of dengue transmission through this mode. For the predictions, four different approaches provide reasonably accurate forecasts. However, some manual calibrations of the tools are needed to make the predictions more accurate at a local scale.

Future improvements include increasing the accuracy of the population abundance estimation in areas where no data are available, exploring the dynamics of female mosquito vertical transmission, optimizing the layout of the webpage, extrapolating the mosquito population dynamics from surveyed areas to unknown regions, and extending the tool to *Culex* mosquito genera.

## Figures and Tables

**Figure 1 pathogens-12-00771-f001:**
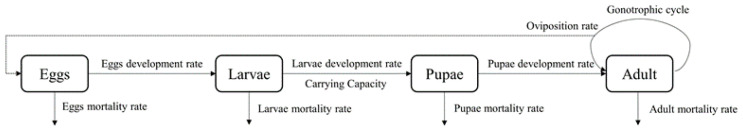
Life cycle of *Aedes aegypti* and the parameters influencing the transition between different stages.

**Figure 2 pathogens-12-00771-f002:**
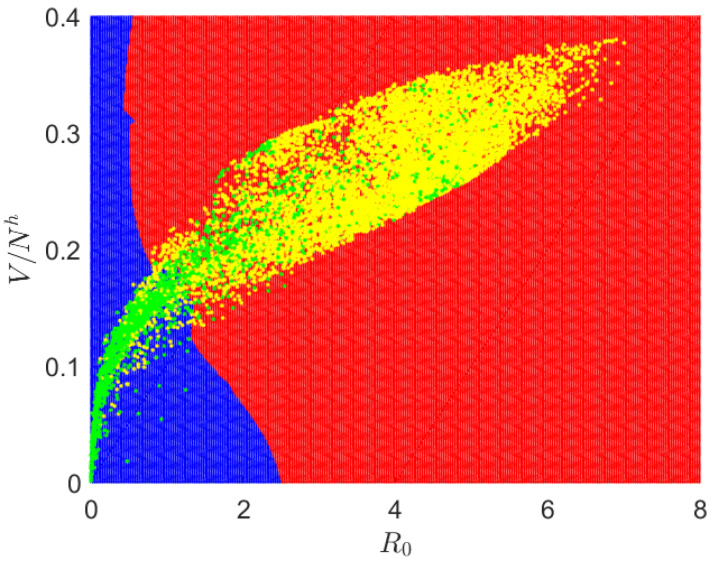
Dengue transmission risk map. Green and yellow dots represent training data; red and blue indicate the risk value above and below 0.5.

**Figure 3 pathogens-12-00771-f003:**
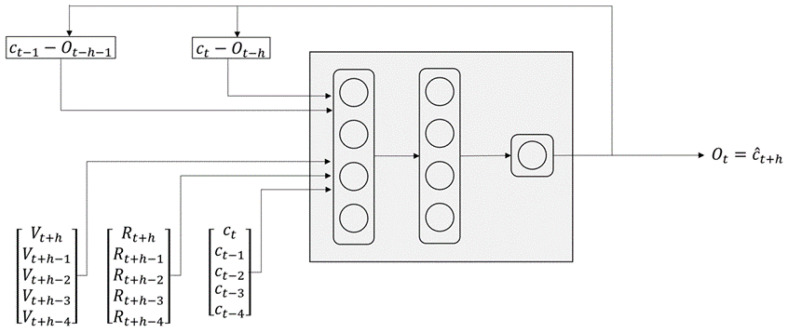
Neural network model for predicting dengue cases.

**Figure 4 pathogens-12-00771-f004:**
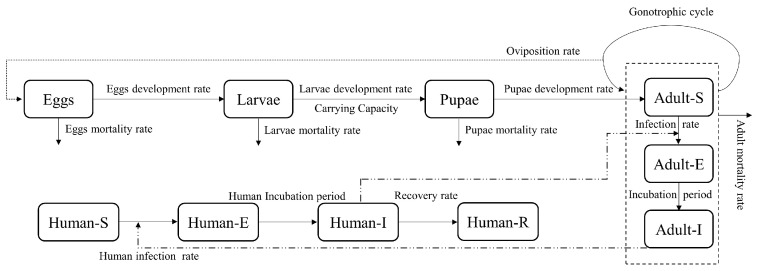
Particle filter transmission model for forecasting dengue cases.

**Figure 5 pathogens-12-00771-f005:**
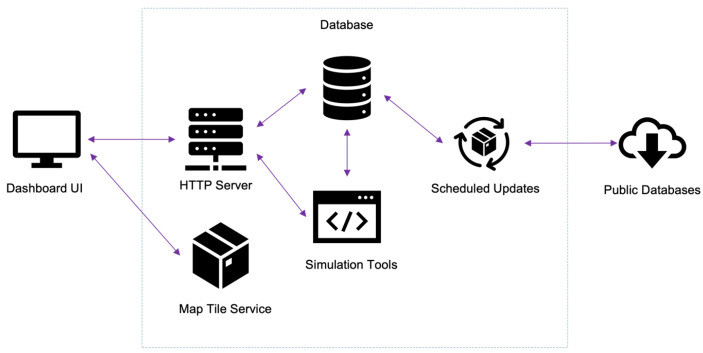
Components and architecture of the application.

**Figure 6 pathogens-12-00771-f006:**
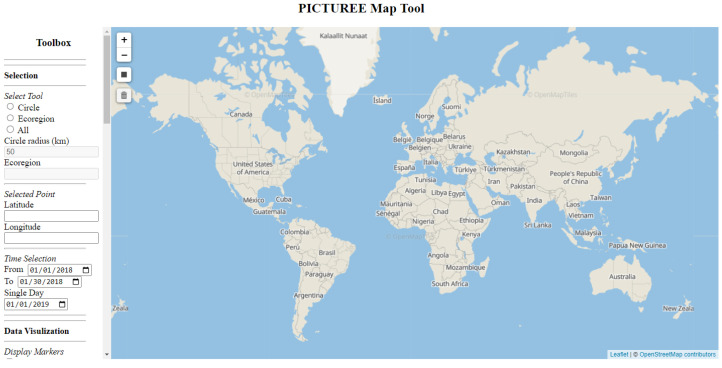
Graphical interface of the PICTUREE–Aedes.

**Figure 7 pathogens-12-00771-f007:**
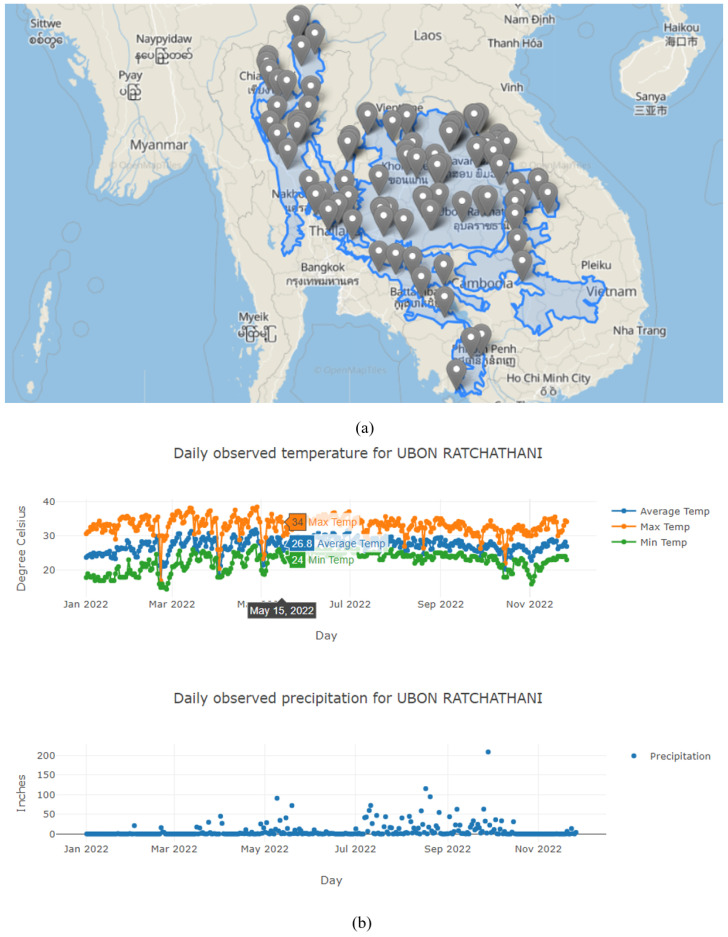
Weather station data. (**a**) Locations of the weather stations in Central Indochina. (**b**) Temperature and precipitation data of the selected station in 2022.

**Figure 8 pathogens-12-00771-f008:**
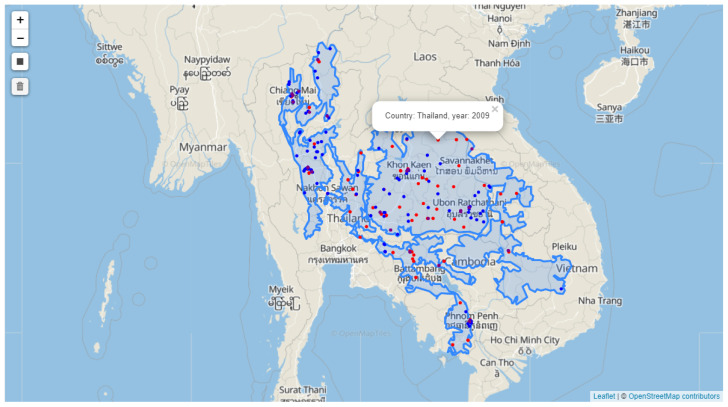
Occurrence data of dengue, *Aedes aegypti* mosquito, and *Aedes albopictus* mosquito.

**Figure 9 pathogens-12-00771-f009:**
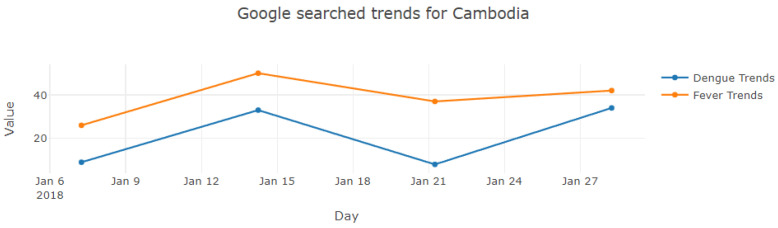
Google search trends of “dengue” and “fever”.

**Figure 10 pathogens-12-00771-f010:**
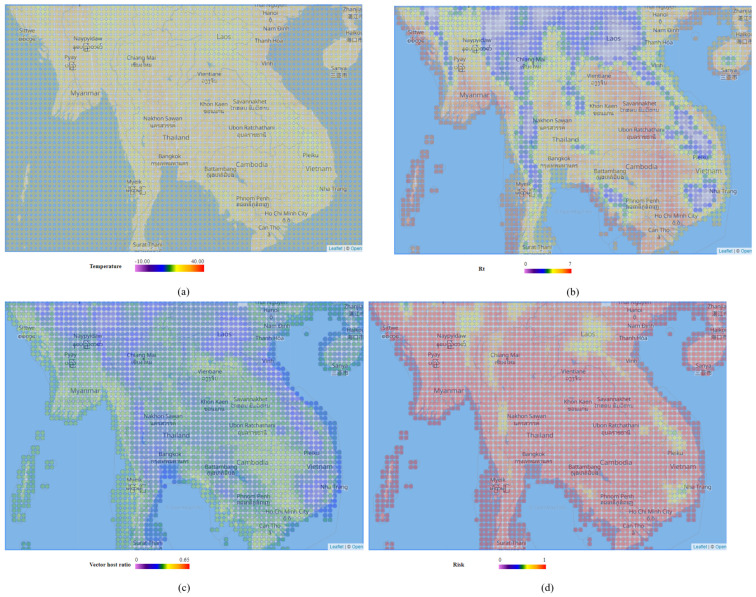
Mosquito simulation results visualized spatially. (**a**) Reanalyzed average temperature, range [−10, 40]. (**b**) Dengue transmission reproduction number $R_{t}$, range [0, 7]. (**c**) Vector–host ratio, range [0, 0.65]. (**d**) Dengue transmission risk, range [0, 1].

**Figure 11 pathogens-12-00771-f011:**
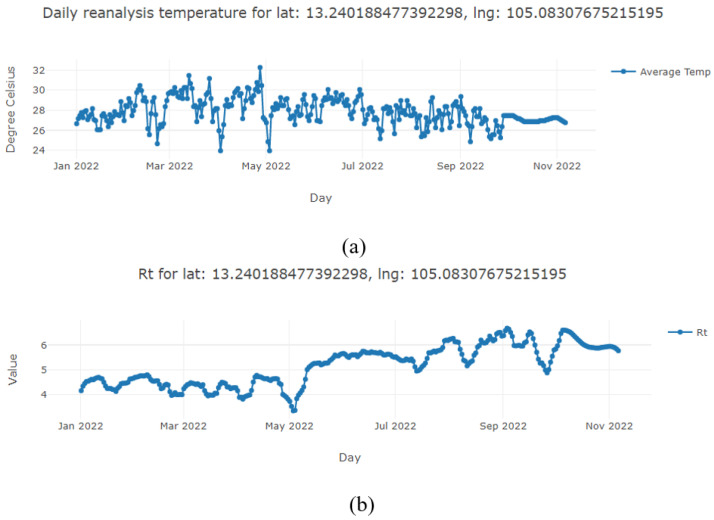
Mosquito simulation results visualized temporally. (**a**) Reanalyzed average temperature. (**b**) Dengue transmission reproduction number. (**c**) Vector–host ratio. (**d**) Dengue transmission risk.

**Figure 12 pathogens-12-00771-f012:**
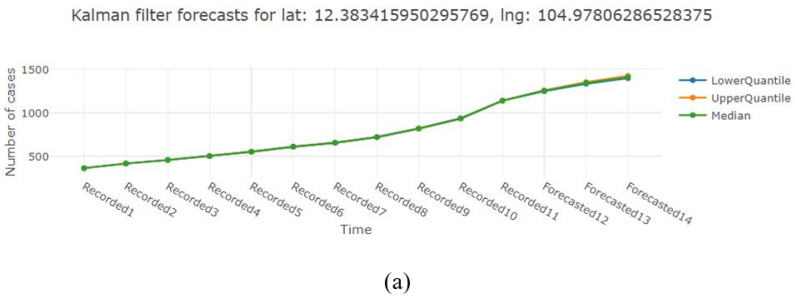
Forecasts of the dengue cases in Cambodia. (**a**) Forecasts of the SEIR-SEI-EnKF model. (**b**) Forecasts of the neural network model. (**c**) Forecasts of the particle filter model. (**d**) Forecasts of the super-ensemble model.

**Table 1 pathogens-12-00771-t001:** Data sources.

Item	Source	Remark
Daily weather observation	[24]	Updated weekly
ERA5 hourly data	[25]	2018–present
Terrestrial ecoregions	[26,27,28]	As of 14 December 2009
*Aedes* mosquito occurrence	[29]	1960–2014
Dengue occurrence	[30]	1960–2012
Country-wise Google trends	[31]	2015–2022

**Table 2 pathogens-12-00771-t002:** Number of predicted cumulative cases and relative absolute error in Cambodia.

	27 April 2022	4 May 2022	11 May 2022
Reported cases	1379	1685	1995
Kalman filter (Median)	1229/10.9%	1293/23.3%	1338/32.9%
Neural network	1414/2.5%	1795/6.5%	2243/12.4%
Particle filter (Median)	1298/5.9%	1492/11.5%	1686/15.5%
Super ensemble (weighted average)	1302/5.6%	1500/11.0%	1703/14.6%

## Data Availability

The data presented in this study are openly available. Weather station data can be found here: https://www.ncei.noaa.gov/access/metadata/landing-page/bin/iso?id=gov.noaa.ncdc:C00516 (accessed on 20 April 2023). ERA5 hourly data are openly available in “ERA5 hourly data on single levels from 1940 to present. Copernicus Climate Change Service (C3S) Climate Data Store (CDS)”, DOI: 10.24381, reference number [25]. *Aedes* mosquito occurrence data are openly available in “Kraemer M. U. G. 2015. Dryad Digital Repository. http://dx.doi.org/10.5061/dryad.47v3c”, reference number [29]. Dengue occurrence data is openly available in “Messina, J., Brady, O., Pigott, D. et al. A global compendium of human dengue virus occurrence. Sci Data 1, 140004 (2014). https://doi.org/10.1038/sdata.2014.4”, reference number [30]. Google trends data can be found here: https://trends.google.com/trends/explore?date=now%201-d&geo=US&q=dengue,fever&hl=en (accessed on 20 April 2023). Other data are available on request from the corresponding author.

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
