# Peer review of "PICTUREE—Aedes: A Web Application for Dengue Data Visualization and Case Prediction"

_pathogens, 2023, doi:10.3390/pathogens12060771_

Round 1

Reviewer 1 Report

he work describes the implementation of various models, various approaches and methodologies to comprehensively address the risk of Dengue and generate a web application. It is then a job more oriented to information technology, informatics or biomedicine. It is very comprehensive in terms of diferent factors influencing Dengue risk. But very complex in the same way.
SMain comments:
in section 3.1 please refer to a table or graph or quote to refer to the calculation of the differential equations of the mechanistic model. in section 3.1.1 How many Dengue Outbrecks were used to adjust or introduce precipitation into the equations? Or provide more detail about the "particle filtering smoothing" methodology with any appointment or work in this area.
in sec. 3.1.3 The authors claim to have used dengue autobreaks around the world. Please see the citations, databases or links to which you accessed for that information. It could also provide a degree of adjustment of the logistic model with which they adjusted the risk of dengue to values ​​between 0 and 1.

The results section is almost entirely based on graphical output from the web application. And the supplementary material is a web server user guide. Figure 6 refers to local meteorological stations, whose intervention in the models is not very clear. If the differential equations take such data, it must be expressed in the legends of the figures or in the text. Because, on the contrary, the authors do speak of re-analysis data for some future calculations. (not from punctual meteorological stations). In figure 8, reference is made to the data obtained from global trends for the month of January 2018, but the text refers to the year 2018. While the data in the figure refer to meteorological seasons of 2022. It would be good to make these figures compatible for the same period of time. Both in the text and in figures 10 and 11 there is a mention to a period of prediction time. And it is also mentioned in the methodology an error cone of the predictions that is not observed in the graphs. The authors should provide some kind of information or justification about the forecast period: 6 months for the variables and 3 or 4 weeks for the risk of Dengue in its different outputs.  

Table 2 should provide some type of estimate or measure of the accuracy of the prediction of each implemented methodology with respect to the actual cases reported. Information or justification about the forecast period: 6 months for the variables and 3 or 4 weeks for the risk of Dengue in its different outputs.
Discussion
It is focus in some biophysicall factors that could could be source of error in the model, but it lucks of references at all.
There is no mention to the accurancy of the predcitions showed, and this could be a central point of this section. On the other hand the authors could focus on the
utility of these types of tools in politicians and decision makers in health issues.

Author Response

Please find our detailed responses to your comments below.

Q1. in section 3.1 please refer to a table or graph or quote to refer to the calculation of the differential equations of the mechanistic model. in section 3.1.1 How many Dengue Outbreaks were used to adjust or introduce precipitation into the equations? Or provide more detail about the "particle filtering smoothing" methodology with any appointment or work in this area.

A1. We shave shown the diagram of the mechanistic model in Figure 1. We added a reference to this figure in section 3.1. However, the detailed mathematical equations are more complex and extensive to be included in this manuscript. They involve theoretical developments that we intend to make available in a companion manuscript currently under preparation.

For the estimation process, we used 67 dengue epidemic curves from Brazil, Mexico, Malaysia, Pakistan, Philippines, Singapore, Taiwan, Sri Lanka, Costa Rica, and Vietnam. Particle filtering is an estimation technique known as sequential Monte Carlo, which involves iteratively updating the estimated parameters using a sequence of observations. We use the number of new cases as observations and the carrying capacity is the unknown parameter that is sequentially estimated.

We have added the references and updated the manuscript accordingly.

Q2. in sec. 3.1.3 The authors claim to have used dengue outbreaks around the world. Please see the citations, databases, or links to which you accessed for that information. It could also provide a degree of adjustment of the logistic model with which they adjusted the risk of dengue to values between 0 and 1.

A2. We used dengue epidemic data from Brazil, Mexico, Malaysia, Pakistan, Philippines, Singapore, Taiwan, Sri Lanka, Costa Rica, and Vietnam. We have updated the manuscript with the proper citations.

After submitting the manuscript, we modified our classification method from a logistic model to a neural network classifier that allows nonlinearity in the classification. We updated the manuscript to reflect this modification. Indeed, this neural network classifier is a generalization of logistic regression which reduce to the logistic model if the neural network has only one hidden layer with only one node. A detailed description of this classifier is becoming available in a companion manuscript currently under preparation. 

In the figure below, we have shown the data used for training as green and yellow dots, indicating low (0) and high (1) transmission, respectively. The neural network classifier is a nonlinear model that assigns a risk value between zero and one to points on the surface defined by the predictive variables V/Nh and R0. Within the depicted image, regions displaying risk values exceeding 0.5 are indicated by the color red, while those with values below 0.5 are represented by the color blue.

Q3. The results section is almost entirely based on graphical output from the web application. And the supplementary material is a web server user guide. Figure 6 refers to local meteorological stations, whose intervention in the models is not very clear. If the differential equations take such data, it must be expressed in the legends of the figures or in the text. Because, on the contrary, the authors do speak of re-analysis data for some future calculations. (not from punctual meteorological stations).

A3. Thanks for your comments. One of the functions of PICTUREE—Aedes is data collection. For our simulation, we opted to utilize reanalysis data; however, we have retained the raw data we collected in case other researchers express interest in utilizing it. At the end of the Data section, we explicitly mention that certain data is presented directly.

Q4.  In figure 8, reference is made to the data obtained from global trends for the month of January 2018, but the text refers to the year 2018. While the data in the figure refer to meteorological seasons of 2022. It would be good to make these figures compatible for the same period of time. Both in the text and in figures 10 and 11 there is a mention to a period of prediction time.

A4. Thanks for your comments. The timeframe "during 2018" has been revised to "during January 2018." The figures included in this paper aim to showcase the collected data, simulated data, and predicted data at various time periods and locations. It is important to note that the simulations and predictions acquire data directly from the database without utilizing the presented data within the application. The intentional selection of visualizations across different locations and time periods serves to demonstrate the feasibility of presenting a comprehensive view of the entire world and a significant time frame.

Q5. And it is also mentioned in the methodology an error cone of the predictions that is not observed in the graphs.

A5. Thanks for your comments. By referring to “the cone of uncertainty”, we intend to give a proper range of the predicted cases. However, in our application, rather than plotting all the predicted curves, we opted to calculate statistics such as the upper quantile, median, and lower quantile values. These statistics effectively convey the range of possibilities. Therefore, we have revised the wording in our manuscript from "the cone of uncertainty" to "statistics including upper quantile, median, and lower quantile values" to accurately describe our approach. 

Q6. The authors should provide some kind of information or justification about the forecast period: 6 months for the variables and 3 or 4 weeks for the risk of Dengue in its different outputs. 

A6. The selection of a 6-month forecast horizon was made without a specific rationale. In fact, the forecast horizon relies on the availability of dependable environmental variables for the future. Presently, we employ the average temperatures from previous years for the corresponding timeframe in the next six months. However, the current values of the calculated variables are adjusted based on the most recent available environmental data. It's important to note that the current levels will influence the future levels of the calculated variable. However, for the dengue case number forecast horizon we chose 3 weeks because human intervention can change the underlying physical transmission process in the long term. We actually have seen this effect for another epidemic such as COVID.

Q7. Table 2 should provide some type of estimate or measure of the accuracy of the prediction of each implemented methodology with respect to the actual cases reported.

A7. Thanks for your comments. We added the relative absolute error to measure the accuracy of each prediction in the manuscript. Now the table is presented as follows:

Discussion

Q8. It is focus in some biophysicall factors that could be source of error in the model, but it lucks of references at all.

A8. Thanks for your comments. We have included the necessary references as suggested.

World Health Organization. Manual on environmental management for mosquito control, with special emphasis on malaria vectors. World Health Organization, 1982.

Barrera, Roberto, Manuel Amador, and Andrew J. MacKay. "Population dynamics of Aedes aegypti and dengue as influenced by weather and human behavior in San Juan, Puerto Rico." PLoS neglected tropical diseases 5, no. 12 (2011): e1378.

Nguyen, Le Anh P., Archie CA Clements, Jason AL Jeffery, Nguyen Thi Yen, Vu Sinh Nam, Gregory Vaughan, Ramon Shinkfield et al. "Abundance and prevalence of Aedes aegypti immatures and relationships with household water storage in rural areas in southern Viet Nam." International Health 3, no. 2 (2011): 115-125.

Q9. There is no mention to the accuracy of the predcitions showed, and this could be a central point of this section. On the other hand, the authors could focus on the utility of these types of tools in politicians and decision-makers in health issues.

A9. Thanks for your suggestion. We have addressed the usage of PICTUREE—Aedes in the manuscript as suggested.  The following is what we modified:

The PICTUREE-Aedes is an innovative and unique system that serves as a valuable resource for both researchers and health officials in their efforts to advance scientific research and effectively manage dengue outbreaks. By collecting and aggregating a comprehensive set of critical factors related to dengue transmission, such as reanalyzed temperature, precipitation, occurrences of Aedes mosquitoes, and dengue cases, the PICTUREE-Aedes provides researchers with a rich dataset for their investigations. This data-driven approach allows researchers to gain deeper insights into the complex dynamics of dengue transmission and aids in the development of more effective prevention and control strategies. In addition to its research-oriented features, the PICTUREE-Aedes offers essential information to health officials, enabling them to proactively prepare and adapt to the changing situation during dengue outbreaks. The system employs sophisticated algorithms to estimate Aedes mosquito abundance, dengue transmission reproduction number, and dengue risk, providing crucial insights into the intensity and geographical spread of the disease. Armed with this knowledge, health officials can make informed decisions about when and where to implement targeted vector control measures, effectively minimizing the risk of dengue transmission and safeguarding public health. Moreover, the PICTUREE-Aedes goes beyond monitoring and estimation by offering valuable forecasting capabilities. By utilizing its advanced modeling techniques, the system can predict future dengue case counts, empowering health officials and healthcare facilities to deploy medical personnel and allocate treatment resources more efficiently. This forecasting functionality is instrumental in anticipating the potential impact of dengue outbreaks, enabling proactive measures to be taken in terms of resource allocation, capacity planning, and public health awareness campaigns.

Thank you once again for your time and effort in reviewing our work.

Reviewer 2 Report

1. I like to say that this is the most complete mathematical epidemiological model for Dengue transmission I have ever seen. Congrats to the authors on this achievement!

2. PICTUREE model integrated many more variables than older approaches, e.g., vector populations and longevity, Dengue Ro transmission, world ecoregions, forecasting, platforms, fancy, advanced math functions, etc.

3. I do not want to demerit this manuscript or disrespect these authors' talent. The local dengue niche is significantly changing in the same city and country. Rain and temperatures may be different from dengue hot spots miles away. However, fine variables, such as the number of susceptible humans and the arrival of new serotypes, are always more sensitive than gross ones. 

4. Three potential adjustments would increase the predictor value of the model results. a) To split adult mosquitos as males and females and then to recalculate predicting cases (males do not bite). 2) To use only the more susceptible populations; e.g., children under nine years old, similar to the very old Vectorial Capacity Malaria model. It could improve results, and 3) Female mosquito vertical transmission through the dry quiescent egg population is increasing its importance, and some authors estimate around 5% of eggs are infected with viruses. They are the ones to trigger dengue incidence after the dry season in many places

4. I noticed that authors did consider some of these observations; I only suggest feeding the discussion section with more comments on how to improve the confidence degree of their model

5. In some paragraphs, the language is very technical, maths, and formulae. Could the authors consider developing concluding sentences adapted for MDs, Entomologists, and epidemiologists?

Author Response

We would like to express our sincere gratitude for your valuable comments, which have greatly assisted us in enhancing the quality of our paper. We have carefully and thoroughly revised the paper to address the feedback provided. Please find our detailed responses to your comments below.

  1. I like to say that this is the most complete mathematical epidemiological model for Dengue transmission I have ever seen. Congrats to the authors on this achievement!
  2. PICTUREE model integrated many more variables than older approaches, e.g., vector populations and longevity, Dengue Ro transmission, world ecoregions, forecasting, platforms, fancy, advanced math functions, etc.

Thank you so much!

Q3. I do not want to demerit this manuscript or disrespect these authors' talent. The local dengue niche is significantly changing in the same city and country. Rain and temperatures may be different from dengue hot spots miles away. However, fine variables, such as the number of susceptible humans and the arrival of new serotypes, are always more sensitive than gross ones.

A3. We appreciate your insightful comment regarding the changing nature of the local dengue niche within the same city and country. We acknowledge that factors such as rain and temperatures can vary significantly even within relatively small distances, which can impact the dynamics of dengue transmission. While it is true that fine variables, such as the number of susceptible humans and the arrival of new serotypes, are highly influential in dengue transmission dynamics, we would like to note that obtaining detailed and comprehensive data on these variables can pose certain challenges. In our study, due to limited data availability, we were unable to incorporate information on specific serotypes and individual movement data between cities and countries. Similarly, obtaining precise temperature and rainfall data for every location is often constrained by data availability and resource limitations. We hope that future research efforts and advancements in data collection methods will further enhance our understanding of the intricate dynamics of dengue transmission, allowing for more precise and localized analyses.

  1. Three potential adjustments would increase the predictor value of the model results. a) To split adult mosquitos as males and females and then to recalculate predicting cases (males do not bite). 2) To use only the more susceptible populations; e.g., children under nine years old, similar to the very old Vectorial Capacity Malaria model. It could improve results, and 3) Female mosquito vertical transmission through the dry quiescent egg population is increasing its importance, and some authors estimate around 5% of eggs are infected with viruses. They are the ones to trigger dengue incidence after the dry season in many places

A4. We appreciate your valuable suggestions to further enhance the predictive value of our model. We address each point below: 1) Regarding the separation of adult mosquitoes into males and females, we would like to clarify that in our simulations, we do consider only female mosquitoes. 2) The reviewer suggests using only more susceptible populations, specifically children under nine years old. While we acknowledge that such an approach could potentially improve the model's accuracy, we unfortunately do not have access to data that allows us to stratify the population in this manner. Estimating the population specifically for this age group and validating such data would introduce complexities and limitations beyond the scope of our current study. 3) We fully agree that female mosquito vertical transmission through the dry quiescent egg population is a significant aspect to consider in our future work. Incorporating the vertical transmission of viruses from infected female mosquitoes to their offspring during the dry season could provide valuable insights into the triggers of dengue incidence following periods of low mosquito activity. We will carefully consider this aspect and explore its implications in our future research.

Q5. I noticed that authors did consider some of these observations; I only suggest feeding the discussion section with more comments on how to improve the confidence degree of their model

A5. We appreciate your suggestions. We agree that considering the factors mentioned before would indeed contribute to a significant enhancement in confidence. However, incorporating these factors into our model requires a sufficient amount of data. In our future work, we will strive to incorporate these factors and their associated data to improve the confidence degree of our model.

Q6. In some paragraphs, the language is very technical, maths, and formulae. Could the authors consider developing concluding sentences adapted for MDs, Entomologists, and epidemiologists?

A6. We appreciate your valuable feedback. We have included the following sentences within the "Simulation tools" section to enhance the readers' comprehension of the methods employed. We hope that these additions aid in providing a clearer understanding of our approach.

At the end of the first paragraph in the Section Simulation tools, we add “The tools graphically represent current and future risk of dengue transmission for users based on the life history of the mosquito vector.”

At the end of the last paragraph in the Subsection Dengue transmission reproductive number, we add “Higher Ro values indicate greater dengue virus transmission in an area, therefore users can quickly understand where control measures need to be initiated and evaluate if they are working by observing changes in Ro.”

At the end of the first paragraph in the Subection Forecasting dengue case number, we add “Case forecasts help PICTUREE users understand how the dengue outbreak will change with time. Forecasts of increasing cases suggest allocating addition resources to an area are needed due to continued transmission, whereas decreasing case forecasts suggest resources can be reallocated and the outbreak is getting smaller.”

At the end of the last paragraph in the Subsection SEIR-SEI-EnKF forecast, we add “In summary, The EnKF utilizes a data assimilation approach that combines real-time observations with mathematical models to accurately forecast dengue epidemics, helping to inform proactive measures for disease control and prevention.”

Round 2

Reviewer 1 Report

I found most of my suggestions done. The manuscript improved significantly. Iḿ only suggest to imporve some graphicall outputs , where axis are not legible.